# Postoperative pneumonia in isolated coronary artery bypass grafting: A comprehensive study of epidemiology, etiology, and disease burden from China (2020–2023)

**Yuxiao Zhan[1], Jian Zhang[2], Yang Yang[1], Rui Yang[3], Guojun Zhang [3]\***

**1** Department of Nosocomial Infection Control, The First Affiliated Hospital of Zhengzhou University, Zhengzhou, Henan, China, **2** Department of Cardiovascular Surgery, The First Affiliated Hospital of Zhengzhou University, Zhengzhou, Henan, China, **3** Department of Respiratory and Critical Care Medicine, The First Affiliated Hospital of Zhengzhou University, Zhengzhou, Henan, China

\* gjzhangzzu@126.com

## Abstract

### Objective

To investigate the epidemiological trends,etiological profiles and disease burden metrics related to postoperative pneumonia following isolated coronary artery bypass grafting (CABG).

### Methods

A retrospective analysis was conducted on data from 518 patients who developed postoperative pneumonia following isolated CABG between January 1, 2020, and November 30, 2023.

### Results

Postoperative pneumonia occurred at a rate of 11.34% among the cohort (518/4569),which fluctuated by year (P < 0.001). Patients aged 60–80 and over 80 years exhibited significantly higher incidence rates compared to those under 60 years (all P < 0.05) A total of 416 strains were identified, with Gram-negative bacteria accounting for 86.5%, primarily represented by *Klebsiella pneumoniae* (31.0%), while *pseudomonas aeruginosa* (21.4%) and *stenotrophomonas maltophilia* (5.3%) demonstrated an increasing trend in the period of 2022–2023 (both P < 0.05). The proportion of *Staphylococcus aureus* in the fourth quarter was significant lower than that in the first quarter (4.8% vs 14.4%, P < 0.05). The overall detection rate of multi-drug resistant organisms (MDRO) was 57.7%,with 53.9% for Gram-negative bacteria and 82.1% for Gram-positive bacteria.Late-onset postoperative pneumonia was significantly associated with a higher detection rate of MDRO (63.8% vs 50.3%, P < 0.01). Postoperative pneumonia prolonged median length of

**Data availability statement:** All relevant data are within the manuscript and its Supporting information files.

**Funding:** The 2nd author (JZ) received a grant from Henan Province medical science and technology research plan joint construction project (LHGJ20210315). The funder had no role in the study design, data collection and analysis, decision to publish, or preparation of the manuscript.

**Competing interests:** All authors declare that there are no conflicts of interest.

postoperative hospital[20.00 (13.00,31.25) days vs 15.50(10.25,19.75) days, $P < 0.001$] and ICU [9.00(5.00,14.00)days vs 4.00(3.00,11.75) days, $P = 0.002$] stay, thereby increasing hospitalization costs[￥255592.15 (193384.29, 336337.53) vs ￥180501.02 (154493.58, 220501.03),$P < 0.001$]. The incidence of severe pneumonia significantly increased in patients infected with MDRO (19.30% vs. 5.51%, $P < 0.001$) or co-infected (40.00% vs. 9.52%, $P < 0.001$), leading to marked differences in postoperative hospital stay and hospitalization costs (all $P < 0.05$).

## Conclusion

The etiological patterns of postoperative pneumonia following isolated CABG showed temporal variations by year and quarter. MDRO infection and co-infections could significantly exacerbate the disease burden on patients.

## Introduction

Coronary Artery Bypass Grafting (CABG) is internationally acknowledged as the most effective treatment for cardiovascular disease and the most common surgical procedure in cardiac surgery. Given the prolonged duration of surgery, extended intubation time, considerable surgical trauma, and complex procedures, hospital-acquired infections have emerged as one of the most frequent postoperative complications for CABG patients. Among these, postoperative pneumonia (POP) is the most predominant type [1–3].In recent years, owing to the evolving spectrum of POP-causing pathogens and the increasingly higher detection rate of muti-drug resistant organisms (MDRO), anti-infective treatment has confronted major challenges. Pathogens are frequently not identified promptly as traditional microbial culture and sensitivity testing are time-consuming.Therefore, the effective selection of early empirical anti-infective drugs is vital for improving outcomes, which relies on dynamic monitoring of the etiology of specialized diseases.Currently, reports on the incidence of POP in CABG patients exhibit substantial variations, ranging from 2.5% to 20.6% [2,4–11], and studies on the disease burden associated with POP are limited, particularly those concentrating on etiological research.The study aims to delineate the epidemiological patterns and etiological profiles of POP in the CABG population, as well as to quantify its impact on disease burden. This research intends to inform the early initiation of empirical anti-infective protocols in clinical practice and to underscore the need for close surveillance of POP which will aid in the development of targeted strategies for prevention and control strategies.

## Subject and methods

### Study design and data collection

The study was conducted as a straightforward retrospective analysis.,which enrolling a total of 5360 patients (aged 18 years and older) who underwent CABG surgery at three of our hospital campuses between January 1, 2020 and November 30, 2023.The data were accessed for this research purpose from December 1, 2023 to April 30, 2024. 4569 patients were selected as study subjects based on inclusion and exclusion criteria, (Fig.1). Exclusion criteria were as follows: a) Patients undergoing combined surgeries. b) Patients receiving invasive mechanical ventilation prior to surgery. c) Patients with a history of chronic lung diseases and not in a stable condition. d) Patients who died or were discharged within 48 hours postoperatively This study was approved by the Research and Clinical Ethics Committee of the First Affiliated

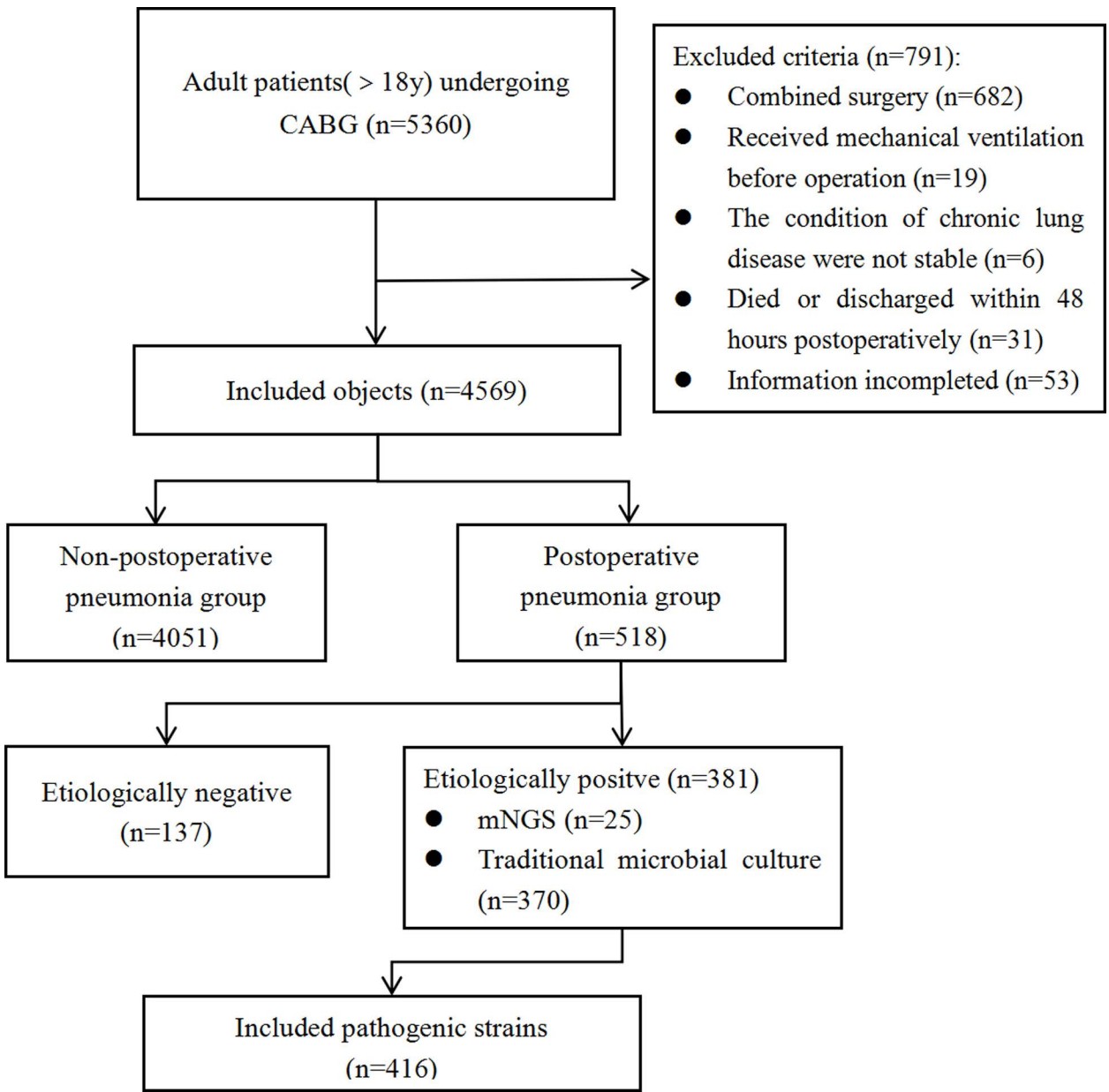

**Fig 1. Data collection flow chart.**

Hospital of Zhengzhou University (2023-KY-1086-001).All patient data had been anonymized during the collection and processing stage, and authors had no access to individual identifying information, we requested an exemption from informed consent.

The collected data encompassed basic patient information, surgical details, diagnostic data across various time frames, microbial test results during hospitalization, ICU length of stay, postoperative hospital stay, and total hospitalization costs.All Data mentioned above had been meticulously gathered through the real-time hospital infection surveillance system.Etiological Data Collection Rules: a)Only respiratory specimen culture results were collected; b)Cultures from inadequate sputum specimens were excluded, defined as sputum smears with >25

squamous epithelial cells per low-power field; c)For the same patient with repeated cultures from the same infection site, only the first strain's susceptibility results were analyzed.

POP patients were divided into MDRO group and non-MDRO group according to whether MDRO was detected. Based on whether two or more species of pathogenic bacteria were detected, they were divided into the co-infection group and simple infection group. According to whether POP occurred within 4 days postoperatively, they were categorized as the early-onset POP group and the late-onset POP group. Note: a)Patients infected with both susceptible bacteria and MDRO were classified into MDR group; b)Patients without etiology were classified into the simple infection group.

## Definitions and diagnostic criteria for postoperative pneumonia and severe pneumonia

Guided by the expert consensus on postoperative pneumonia in China [12],"postoperative pneumonia" was hereby delineated to include pneumonia identified within the 30-day window after surgery.

Based on the diagnostic criteria for pneumonia (WS 382-2012) and the expert consensus on postoperative pneumonia in China [12], POP was diagnosed when criteria A, B, and C were met concurrently,which was aligned with the pneumonia definition updated by the U.S. NHSN in January 2024:

A. At least two of the following: a) New or changing purulent sputum or increased respiratory secretions, or more frequently suctioning; b) New onset of cough, dyspnea, or increased respiratory rate, or worsened pre-existing cough, dyspnea, or shortness of breath; c) New or worsened lung crackles or bronchial breath sounds; d) Deterioration in gas exchange, increased oxygen requirements, or need for mechanical ventilation support.

B. At least one of the following: a) Fever (T > 38 °C) with no other obvious cause; b) Peripheral blood WBC count > $12 \times 10^9$/L or < $4 \times 10^9$/L; c) Altered mental status in individuals aged ≥ 70 years with no other obvious cause.

C. At least two chest X-rays (for patients without underlying heart or lung disease, one chest X-ray may be sufficient), and at least one of the following: a) New or progressive and persistent pulmonary infiltrates; b) Consolidation; c) Cavity formation.

Patients with POP can be classified as having severe pneumonia if they exhibit at least one of the following criteria [13]: a)Requirement for intubation and mechanical ventilation; b) Occurrence of septic shock requiring vasopressor therapy despite aggressive fluid resuscitation.

## Strain identification and antimicrobial susceptibility testing

Bacterial strain identification and antimicrobial susceptibility testing (MIC determination) were performed with the VITEK 2 Compact system. For pathogens with undetermined MIC values by the instrument, the K-B disk diffusion method was employed for supplementary test. The susceptibility results were interpreted according to the 2020 Clinical and Laboratory Standards Institute (CLSI) M100 standards (M100-S30).

## Statistical analysis

To mitigate the impact of confounding factors on our study outcomes, we meticulously controlled the inclusion and exclusion criteria during the study design phase. We rigorously screened the data. Given that the missing data exhibited a pattern of random absence, with a missing proportion less than 10%, we treated these missing values as blank entries.

Data were processed via SPSS version 22.0. The Kolmogorov-Smirnov test was used to assess the normality of the distribution of quantitative variables. Non-normally distributed continuous data were presented as median (P25–P75), and differences among groups were compared with the two-tailed Mann-Whitney U test. Categorical data were described using frequencies and percentages, and differences among groups were compared through the $\chi^2$ test. Multiple linear regression analysis was employed to adjust for potentially confounding factors. A p-value of < 0.05 was considered statistically significant.

## Results

### Epidemiology

Among the 4569 patients, 518 developed POP, with an overall incidence of 11.34%. Patients with POP included 379 males (73.17%) and 139 females (26.83%), with an average age of 63.82 ± 8.80 years.The incidence of POP in emergency surgeries was significantly higher, reaching 18.75% (15/80),compared to the non-emergency cohort with a rate of 11.21% (503/4489) (P = 0.035).When stratified by sex, the incidence of POP was 11.69% (379/3243) in males and 10.48% (139/1326) in females, with no statistically significant difference observed (P = 0.244). Furthermore, age-specific incidence rates revealed a gradient effect, with patients under 60 years experiencing an incidence of 8.72% (159/1824), those aged 60 to 80 years showing a higher rate of 12.90% (351/2720), and those over 80 years old having the highest incidence at 25.00% (8/32).Compared to the incidence among patients under 60 years, those aged 60 to 80 and over 80 years exhibited significantly higher rates (both P < 0.05).The annual incidences of POP from 2020 to 2023 were 9.80% (80/816), 8.69% (107/1232), 11.91% (113/949), and 13.87% (218/1572), respectively.Notably, the interannual variations are significantly different (2022 vs 2021,2023 vs2020,2023 vs2021,all P < 0.05). The incidence rates of POP across the quarters were as follows: 12.26% (163/1330) in the first quarter, 11.06% (139/1257) in the second quarter, 9.70% (103/1062) in the third quarter, and 12.28% (113/920) in the fourth quarter. Across blood types A, B, O, and AB, the incidence rates were 12.22% (173/1416), 11.48% (165/1437), 10.91% (133/1219), and 9.46% (47/497), respectively.While statistical analysis revealed no significant differences in incidence among the different quarters and blood types (all P > 0.05) (Table 1).

### Etiology

**Distribution of pathogens.**  A total of 416 pathogen strains were isolated from 381 patients with positive etiological tests. Among these, 360 strains (86.5%) were Gram-negative bacteria, and 56 strains (13.5%) were Gram-positive bacteria. Among the 416 pathogenic strains, 240 were MDRO, yielding an overall detection rate of 57.7%. with 53.9% for Gram-negative bacteria and 82.1% for Gram-positive bacteria (Table 2).The top five pathogens were *Klebsiella pneumoniae* (31.0%, 129 strains), *Pseudomonas aeruginosa* (21.4%, 89 strains), *Acinetobacter baumannii* (16.8%, 70 strains), *Staphylococcus aureus* (9.6%, 40 strains), and *Stenotrophomonas maltophilia* (5.3%, 22 strains) (Fig 2).

### Variations in the distribution and detection of MDRO under diverse circumstances

**Variations over the years.**  Compared with the period of 2020–2021, the proportion of *Enterobacteriaceae* declined, while that of non-fermentative bacteria rose in 2022–2023. Notably, the increases in *P.aeruginosa* and *S.maltophilia* were significant (both P < 0.05). No statistically significant variations were observed in the distribution of other bacterial species between the two periods (all P > 0.05).The detection rates of MDRO for various species among different periods showed no substantial distinctions (all P > 0.05) (Table 2).

**Table 1. Analysis of epidemiological characteristics of postoperative pneumonia.**

| Variables | POP Group (n = 518) | non-POP Group (n = 4051) | P value* |
|---|---|---|---|
| **Type of surgery** | | | |
| Emergency surgery | 15 (2.90%) | 65 (1.60%) | 0.035 |
| Non-emergency surgery | 503 (97.10%) | 3986 (98.40%) | |
| **Gender** | | | |
| Male | 379 (73.17%) | 2864 (70.70%) | 0.244 |
| Female | 139 (26.83%) | 1187 (29.30%) | |
| **Age** | | | |
| <60y | 159 (30.69%)[a] | 1665 (41.10%) | <0.001 |
| 60–80y | 351 (67.76%)[bc] | 2369 (58.48%) | |
| >80y | 8 (1.54%)[c] | 24 (0.59%) | |
| **Year** | | | |
| 2020 | 80 (15.44%)[ac] | 736 (18.17%) | <0.001 |
| 2021 | 107 (20.66%)[a] | 1125 (27.77%) | |
| 2022 | 113 (21.81%)[bc] | 836 (20.64%) | |
| 2023 | 218 (42.08%)[b] | 1354 (33.42%) | |
| **Quarter** | | | |
| Q1 | 163 (31.47%) | 1167 (28.81%) | 0.182 |
| Q2 | 139 (26.83%) | 1118 (27.60%) | |
| Q3 | 103 (19.88%) | 959 (23.67%) | |
| Q4 | 113 (21.81%) | 807 (19.92%) | |
| **Blood type** | | | |
| A | 173 (33.40%) | 1243 (30.68%) | 0.378 |
| B | 165 (31.86%) | 1272 (31.40%) | |
| O | 133 (25.68%) | 1086 (26.81%) | |
| AB | 47 (9.07%) | 450 (11.11%) | |

*P values

were calculated using the $\chi^2$ test. Different letters (a, b, c) denoted statistically significant differences between groups (P < 0.05), whereas containing the same letter indicated no significant difference (P > 0.05). For example, the presence of different letters such as "a" and "b," or "a" and "bc" signify a significant difference between groups, while "a" and "ac" indicates no significant difference.

The same letters between groups indicated no statistical difference (P > 0.05), while different letters indicated statistically significant difference (P < 0.05).

**Variations among the quarters.** The detection of Gram-negative and Gram-positive bacteria fluctuated across quarters. The former exhibited a gradual increase,while the latter presented a gradual decline. Notably, the proportion of Gram-negative and Gram-positive bacteria in the fourth quarter (Q4) showed significant differences compared to the first (Q1) and second (Q2) quarters (all P < 0.05). Among Gram-positive bacteria, the proportion of *S.aureus* in Q4 differed significantly from that in Q1. No other differences between quarters were statistically significant (all P > 0.05). Furthermore, no substantial distinctions were observed in the detection rates of MDRO for various species among different quarters (all P > 0.05) (Table 3).

**Variations in the early-onset group and late-onset group.** A comparative analysis of pathogen distribution between the early-onset and late-onset group revealed that the overall composition of pathogens were comparable, with no substantial differences (all P > 0.05). While The late-onset POP group showed a higher overall detection rate for MDRO (P < 0.01), particularlly Enterobacteriaceae in Gram-negative bacteria (P < 0.05), compared to those in the early-onset POP group, (Table 4).

**Table 2. The distribution of common pathogens and the corresponding detection rate of MDRO:2020–2021 versus 2022-2023.**

| Bacterial Species | Total | | 2020–2021 | | 2022–2023 | |
|---|---|---|---|---|---|---|
| | Strains (%) | MDR-Strains (%) | Strains (%) | MDR-Strains (%) | Strains (%) | MDR-Strains (%) |
| **Gram-negative bacteria** | 360 (86.5) | 194 (53.9) | 124 (85.5) | 70 (56.5) | 236 (87.1) | 124 (52.5) |
| *Enterobacteriaceae* | 156 (37.5) | 93 (59.6) | 64 (44.1) | 37 (57.8) | 92 (33.9)* | 56 (60.9) |
| *K.pneumoniae* | 129 (31.0) | 72 (55.8) | 51 (35.2) | 28 (54.9) | 78 (28.8) | 44 (56.4) |
| **Non-fernebtative bacteria** | 196 (47.1) | 99 (50.5) | 56 (38.6) | 32 (57.1) | 140 (51.7)* | 67 (47.9) |
| *P.aeruginosa* | 89 (21.4) | 44 (49.4) | 21 (14.5) | 10 (47.6) | 68 (25.1)* | 34 (50.0) |
| *A.baumannii* | 70 (16.8) | 47 (67.1) | 28 (19.3) | 19 (67.9) | 42 (15.5) | 28 (66.7) |
| *S.maltophilia* | 22 (5.3) | 0 (0) | 3 (2.1) | 0 (0) | 19 (7.0)* | 0 (0) |
| **Gram-positive bacteria** | 56 (13.5) | 46 (82.1) | 21 (14.5) | 16 (76.2) | 35 (12.9) | 30 (85.7) |
| *S.aureus* | 40 (9.6) | 36 (90.0) | 15 (10.3) | 14 (93.3) | 25 (9.2) | 22 (88.0) |
| *S.pneumoniae* | 14 (3.4) | 9 (64.3) | 6 (4.1) | 2 (33.3) | 8 (3.0) | 7 (87.5) |
| **Total** | 416 (100) | 240 (57.7) | 145 (100) | 86 (59.3) | 271 (100) | 154 (56.8) |

P values were calculated using the $\chi^2$ test

Compared with 2020–2021, * P < 0.05

## Antimicrobial resistance of common pathogens

**Resistance patterns of common Gram-negative bacteria.** *K.pneumoniae* showed resistance rates of 33.3% to imipenem and 31.9% to meropenem, respectively. It was 100% resistant to ampicillin, with other notable resistance rates to ceftriaxone (58.3%), cefazolin (56.8%), and ampicillin-sulbactam (47.1%), resistance rates to tigecycline (7.7%) and colistin (3.4%) were relatively low. *P.aeruginosa* exhibited resistance rates of 46.0% to imipenem and 31.0% to meropenem, respectively. The resistance rate to piperacillin-clavulanate was 28.9%, with resistance rates to other commonly used antibiotics generally below 20%. The lowest resistance rate was observed for tobramycin (1.1%). *A.baumannii* showed high resistance rates of 62.7% to both imipenem and meropenem. The highest resistance rate was observed for piperacillin-clavulanate (74.0%), followed by ciprofloxacin (70.9%), piperacillin-tazobactam (69.5%), and ampicillin-sulbactam (68.4%). Tigecycline had a lower resistance rate (5.0%), and no colistin-resistant strains were detected.

Compared to the period of 2020–2021, the resistance rates of *K.pneumoniae* to 14 commonly used antibiotics increased in 2022–2023 (14/21). Notably, resistance rates to ciprofloxacin and ceftriaxone increased by over 10%. For *P.aeruginosa*, resistance rates to 7 commonly used antibiotics increased (7/13), with resistance to piperacillin-clavulanate and aztreonam increasing by more than 10%. *A.baumannii* showed increased resistance rates to 3 commonly used antibiotics (3/17), yet the overall rise in resistance to all three antibiotics did not surpass 10%. (Table 5).

**Resistance patterns of common Gram-positive bacteria.** *S.aureus* exhibited a resistance rate of 69.2% to methicillin. The highest resistance was seen in erythromycin (97.4%), followed by clindamycin (91.4%), penicillin G (83.3%), and tetracycline (66.7%). Resistance rates to quinupristin-dalfopristin (6.3%) and rifampin (2.6%) were relatively low. No vancomycin or linezolid-resistant strains were identified.*S. pneumoniae* showed 100% resistance to erythromycin, with other notable resistance rates to tetracycline (88.9%), clindamycin (76.9%), and trimethoprim-sulfamethoxazole (60.0%). No resistance was observed for levofloxacin, moxifloxacin, vancomycin, or linezolid.

Compared to the period of 2020–2021, *S.aureus* showed increased resistance rates to 3 commonly used antibiotics in 2022–2023 (3/14). Specifically, the resistance to

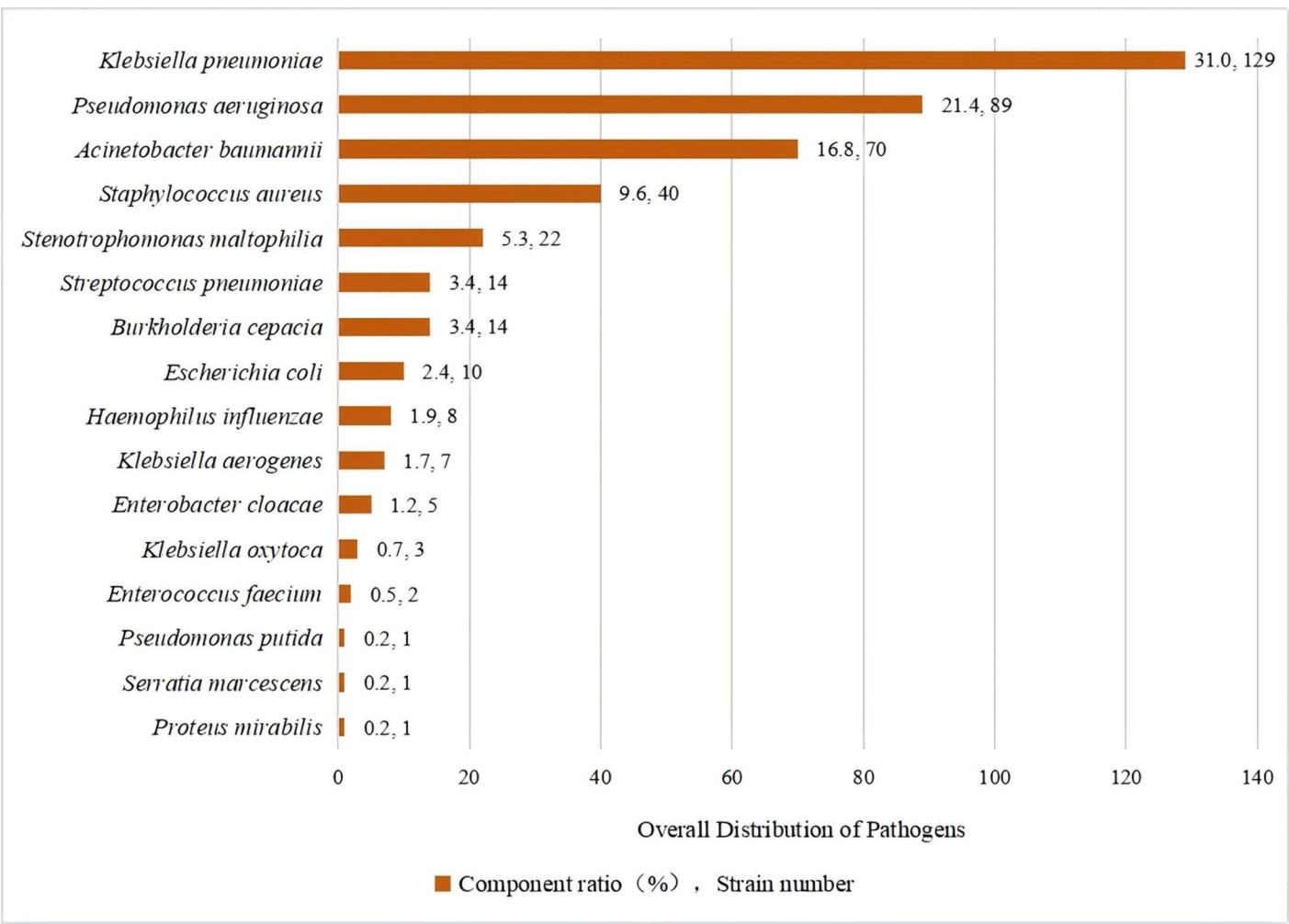

**Fig 2. The overall distribution of pathogens responsible for postoperative pneumonia.**

trimethoprim-sulfamethoxazole increased by more than 30%,and more than 20% to levoflox-acin (Table 6). Due to the small number of strains, *S.pneumoniae* was not encompassed in this analysis.

## Analysis of disease burden related to POP

Patients with MDRO infection (19.30% vs 5.51%,P < 0.001) and co-infection (40.00% vs 9.52%, P < 0.001) exhibited significantly increased risk of developing severe pneumonia, which in turn exacerbated the disease burden. To elucidate the precise impact of these conditions on disease burden, we conducted a subgroup analysis focusing on hospital length of stay and hospitalization costs.

**Length of Hospital stay.** Patients with POP experienced a significantly longer median postoperative hospital stay of 4.50-day compared to those without POP [20.00 (13.00,31.25) days vs 15.50 (10.25,19.75) days, P < 0.001]. The median length of postoperative hospital stay among patients infected with MDRO[22.00 (13.00,34.00) days vs 17.00 (12.00,25.00) days, P = 0.021] and co-infected [26.00 (16.00,35.00) days vs 20.00 (13.00,31.00) days, P < 0.001]were significantly longer than those without the aforementioned conditions. Patients

**Table 3. The distribution of common pathogens and the corresponding detection rate of MDRO:from the first quarter to the fourth quarter.**

| Bacterial Species | The first quarter | | The second quarter | | The Third quarter | | The fourth quarter | |
|---|---|---|---|---|---|---|---|---|
| | Strains (%) | MDR-Strains (%) | Strains (%) | MDR-Strains (%) | Strains (%) | MDR-Strains (%) | Strains (%) | MDR-Strains (%) |
| **Gram-negative bacteria** | 94 (79.7)[a] | 49 (52.1) | 112 (86.2)[a] | 66 (58.9) | 74 (88.1)[ab] | 37 (50.0) | 80 (95.2)[bc] | 42 (52.5) |
| *Enterobacteriaceae* | 42 (35.6) | 24 (57.1) | 51 (39.2) | 35 (68.6) | 30 (35.7) | 18 (60.0) | 33 (39.3) | 16 (48.5) |
| *K.pneumoniae* | 32 (27.1) | 17 (53.1) | 42 (32.3) | 27 (64.3) | 28 (33.3) | 16 (57.1) | 27 (32.1) | 12 (44.4) |
| **Non-fernebtative bacteria** | 50 (42.4) | 24 (48.0) | 59 (45.4) | 31 (52.5) | 42 (50.0) | 19 (45.2) | 45 (53.6) | 25 (55.6) |
| *P.aeruginosa* | 22 (18.6) | 12 (54.5) | 22 (16.9) | 12 (54.5) | 24 (28.6) | 10 (41.7) | 21 (25.0) | 10 (47.6) |
| *A.baumannii* | 18 (15.3) | 11 (61.1) | 25 (19.2) | 18 (72.0) | 9 (10.7) | 7 (77.8) | 18 (21.4) | 11 (61.1) |
| *S.maltophilia* | 7 (5.9) | 0 (0) | 9 (6.9) | 0 (0) | 4 (4.8) | 0 (0) | 2 (2.4) | 0 (0) |
| **Gram-positive bacteria** | 24 (20.3)[a] | 19 (79.2) | 18 (13.8)[a] | 14 (77.8) | 10 (11.9)[ab] | 9 (90) | 4 (4.8)[bc] | 4 (100) |
| *S.aureus* | 17 (14.4)[a] | 16 (94.1) | 12 (9.2)[ab] | 9 (75.0) | 7 (8.3)[ab] | 7 (100) | 4 (4.8)[b] | 4 (100) |
| *S.pneumoniae* | 6 (5.1) | 3 (50.0) | 5 (3.8) | 4 (80.0) | 3 (3.6) | 2 (66.7) | 0 (0) | 0 (0) |
| **Total** | 118 (100) | 68 (57.6) | 130 (100) | 80 (61.5) | 84 (100) | 46 (54.8) | 84 (100) | 46 (54.8) |

P values were calculated using the $\chi^2$ test. Different letters (a, b, c) denoted statistically significant differences between groups (P < 0.05), whereas containing the same letter indicated no significant difference (P > 0.05). For example, the presence of different letters such as "a" and "b," or "a" and "bc" signify a significant difference between groups, while "a" and "ac" indicates no significant difference.

The same letters among groups indicated no statistical difference (P > 0.05), while different letters indicated statistically significant difference (P < 0.05).

**Table 4. The distribution of common pathogens and the corresponding detection rate of MDRO: early-onset group versus late-onset group.**

| Bacterial Species | Early-onset group (≤4d) | | Late-onset group (>5d) | |
|---|---|---|---|---|
| | Strains (%) | MDR-Strains (%) | Strains (%) | MDR-Strains (%) |
| **Gram-negative bacteria** | 160 (85.6) | 73 (45.6) | 200 (87.3) | 121 (60.5)▲ |
| *Enterobacteriaceae* | 69 (36.9) | 34 (49.3) | 87 (38.0) | 59 (67.8)* |
| *K.pneumoniae* | 56 (29.9) | 27 (48.2) | 73 (31.9) | 45 (61.6) |
| **Non-fernebtative bacteria** | 87 (46.5) | 38 (43.7) | 109 (47.6) | 61 (56.0) |
| *P.aeruginosa* | 40 (21.4) | 15 (37.5) | 49 (21.4) | 29 (59.2) |
| *A.baumannii* | 31 (16.6) | 20 (64.5) | 39 (17.0) | 27 (69.2) |
| *S.maltophilia* | 7 (3.7) | 0 (0.0) | 15 (6.6) | 0 (0.0) |
| **Gram-positive bacteria** | 27 (14.4) | 21 (77.8) | 29 (12.7) | 25 (86.2) |
| *S.aureus* | 19 (10.2) | 18 (94.7) | 21 (9.2) | 18 (85.7) |
| *S.pneumoniae* | 6 (3.2) | 2 (33.3) | 8 (3.5) | 7 (87.5) |
| **Total** | 187 (100) | 94 (50.3) | 229 (100) | 146 (63.8)▲ |

P values were calculated using the $\chi^2$ test.

Compared with the early-onset

group, ▲P < 0.01, * P < 0.05.

developed POP had a 5.00-day longer median ICU stay than those who did not [9.00 (5.00,14.00) days vs 4.00 (3.00,11.75) days, P = 0.002], those infected with MDRO had a 4.00-day longer median ICU stay than patients without MDRO infection[11.00 (7.00,16.00) days vs 7.00 (3.00,9.00) days, P < 0.001]. However, there was no statistically significant difference in ICU stay between co-infection group and simple infection group [10.00 (7.00,16.00) days vs 9.00 (4.00,14.00) days, P = 0.189] (Table 7).

## Hospitalization costs

The median hospitalization costs for POP patients increased by ￥75091.13 compared to those without POP[￥255592.15 (193384.29, 336337.53) vs ￥180501.02 (154493.58, 220501.03), P

**Table 5. Resistance rates of common Gram-negative bacteria to antibiotics:2020–2021 versus 2022-2023.**

| Antibiotics | K.pneumoniae | | | | P.aeruginosa | | | | A.baumannii | | | |
|---|---|---|---|---|---|---|---|---|---|---|---|---|
| | 2020–2021 | | 2022–2023 | | 2020–2021 | | 2022–2023 | | 2020–2021 | | 2022–2023 | |
| | Test (Num) | R[n (%)] | Test (Num) | R[n (%)] | Test (Num) | R[n (%)] | Test (Num) | R[n (%)] | Test (Num) | R[n (%)] | Test (Num) | R[n (%)] |
| Ticarcillin-clavulnate | 38 | 15 (39.5) | 72 | 25 (34.7) | 12 | 2 (16.7) | 33 | 11 (33.3) | 21 | 16 (76.2) | 29 | 21 (72.4) |
| Ciprofloxacin | 43 | 14 (32.6) | 69 | 31 (44.9) | 17 | 4 (23.5) | 63 | 11 (17.5) | 23 | 17 (73.9) | 32 | 22 (68.8) |
| Levofloxacin | 48 | 15 (31.3) | 74 | 24 (32.4) | 21 | 6 (28.6) | 66 | 11 (16.7) | 24 | 11 (45.8) | 34 | 17 (50.0) |
| Cefperazone-Sulbactam | 47 | 17 (36.2) | 73 | 24 (32.9) | 14 | 1 (7.1) | 33 | 4 (12.1) | 24 | 13 (54.2) | 35 | 14 (40.0) |
| Trimethoprim-sulfamethoxazole | 48 | 17 (35.4) | 74 | 26 (35.1) | / | / | / | / | 24 | 6 (25.0) | 35 | 8 (22.9) |
| Ceftazidime | 48 | 13 (27.1) | 75 | 24 (32.0) | 21 | 1 (4.8) | 66 | 6 (9.1) | 24 | 15 (62.5) | 34 | 21 (61.8) |
| Olistin | 43 | 0 (0) | 74 | 4 (5.4) | 12 | 0 (0) | 33 | 1 (3.0) | 23 | 0 (0) | 33 | 0 (0) |
| Piperacillin-tazobactam | 48 | 18 (37.5) | 75 | 25 (33.3) | 21 | 2 (9.5) | 65 | 7 (10.8) | 24 | 17 (70.8) | 35 | 24 (68.6) |
| Minocycline | 45 | 11 (24.4) | 73 | 16 (21.9) | / | / | / | / | 23 | 1 (4.3) | 35 | 4 (11.4) |
| Imipenem | 48 | 15 (31.3) | 75 | 26 (34.7) | 21 | 10 (47.6) | 66 | 30 (45.5) | 27 | 17 (63.0) | 40 | 25 (62.5) |
| Cefepime | 47 | 16 (34.0) | 75 | 31 (41.3) | 21 | 1 (4.8) | 65 | 0 (0) | 24 | 13 (54.2) | 35 | 18 (51.4) |
| amikacin | 48 | 10 (20.8) | 75 | 21 (28.0) | 21 | 1 (4.8) | 66 | 4 (6.1) | 24 | 14 (58.3) | 34 | 18 (52.9) |
| Meropenem | 45 | 14 (31.1) | 74 | 24 (32.4) | 21 | 7 (33.3) | 66 | 20 (30.3) | 27 | 17 (63.0) | 40 | 25 (62.5) |
| Doxycycline | 40 | 18 (45.0) | 68 | 26 (38.2) | – | – | – | – | 23 | 15 (65.2) | 31 | 17 (54.8) |
| Ampicillin-Sulbactam | 48 | 20 (41.7) | 71 | 36 (50.7) | – | – | – | – | 23 | 17 (73.9) | 34 | 22 (64.7) |
| Tobramycin | 48 | 10 (20.8) | 73 | 19 (26.0) | 21 | 1 (4.8) | 66 | 0 (0) | 24 | 14 (58.3) | 35 | 18 (51.4) |
| Tigecycline | 45 | 3 (6.7) | 72 | 6 (8.3) | / | / | / | / | 24 | 0 (0) | 35 | 3 (8.6) |
| Aztreonam | 47 | 15 (31.9) | 75 | 30 (40.0) | 21 | 3 (14.3) | 64 | 15 (23.4) | – | – | – | – |
| Ampicillin | 35 | 35 (100) | 65 | 65 (100) | / | / | / | / | / | / | / | / |
| Cefazolin | 44 | 23 (52.3) | 67 | 40 (59.7) | / | / | / | / | / | / | / | / |
| Ceftriaxone | 34 | 18 (52.9) | 26 | 17 (65.4) | / | / | / | / | / | / | / | / |

- Indicates untested,/ indicates natural resistance.

< 0.001]. Among POP patients, those infected with MDRO faced a higher median hospitalization costs than those without MDRO[￥273137.78 (204265.65, 337002.90) vs ￥224766.03 (185254.94, 315148.71), P < 0.001]. Not surprisingly, patients with co-infection also exhibited increased hospitalization costs compared to those with simple infection[￥285400.82 (255684.40, 330173.81) vs ￥242174.39 (191320.92, 336115.74), P < 0.001] (Table 7).

## Discussion

This study found an overall incidence of POP following CABG to be 11.34%. The rate is higher than that presented in the data from international studies [2,5,8,9], but lower than domestic research [7,11]. The discrepancy may result from: a) dissimilar inclusion criteria for study subjects; b) diverse diagnostic standards in POP; c) differences in demographic characteristics of patients across hospitals; d) disparities in baseline risk factors;e) potential systematic biases in monitoring. In this study, patients undergoing emergency surgeries exhibited a heightened incidence of POP, consistent with the findings reported by Xiang et al [14]. Kilic A et al. [15] identified a 1.4-fold increase risk of POP in patients over 65 years of age following cardiac surgery. Similarly, Ailawadi et al. [16] reported a positive correlation between advancing age and the risk of POP in the same context. This study corroborated these findings, indicating that patients aged 60–80 years exhibit a higher susceptibility to POP in comparison with those under 60 years. Although the incidence of POP in patients over 80 years did

**Table 6. Resistance rates of common Gram-positive bacteria to antibiotics:2020–2021 versus 2022-2023.**

| Antibiotics | S.aureus | | | |
|---|---|---|---|---|
| | 2020–2021 | | 2022–2023 | |
| | Test (Num) | R[n (%)] | Test (Num) | R[n (%)] |
| Levofloxacin | 14 | 6 (42.9) | 25 | 16 (64.0) |
| Tetracycline | 14 | 10 (71.4) | 25 | 16 (64.0) |
| Oxacillin | 14 | 11 (78.6) | 25 | 16 (64.0) |
| Vancomycin | 14 | 0 (0) | 25 | 0 (0) |
| Gentamicin | 14 | 6 (42.9) | 25 | 12 (48.0) |
| Penamecillin | 13 | 13 (100) | 17 | 12 (70.6) |
| Trimethoprim-sulfamethoxazole | 12 | 1 (8.3) | 24 | 10 (41.7) |
| Ciprofloxacin | 10 | 6 (60.0) | 17 | 7 (41.2) |
| Clindamycin | 10 | 10 (100) | 25 | 22 (88.0) |
| Moxifloxacin | 14 | 5 (35.7) | 24 | 6 (25.0) |
| Rifampin | 13 | 1 (7.7) | 25 | 0 (0) |
| Linezolid | 14 | 0 (0) | 20 | 0 (0) |
| Erythromycin | 14 | 14 (100) | 24 | 23 (95.8) |
| Quinupristin-Dalfopristin Syncercid | 13 | 2 (15.4) | 19 | 0 (0) |

not demonstrate a statistically significant difference relative to other age groups, there was an observable upward trend, likely attributable to the smaller sample size in this cohort. The incidence of POP was lowest in 2021, then with a subsequent annual increase. This variation might be linked to differences in patient demographics, underlying health conditions, nutritional status, and the effectiveness of POP infection control strategies. Furthermore, this study revealed that sex, the surgical quarter, and blood types had no significant influence on the occurrence of POP.

Existing studies on the pathogen distribution of POP following cardiac surgery demonstrated significant discrepancies [6,17], different study populations and regional variations might explain the wide range in reported data. While researches on pathogen distribution specific to CABG procedures was limited, our study offered insights into the pathogen profile for POP patients post- CABG. We found that Gram-negative bacteria predominated, constituting 86.5% of the isolates, with *K.pneumoniae* and *P.aeruginosa* being the most prevalent. Gram-positive bacteria were identified less frequently, accounting for 13.5%, with *S.aureus* being the most common. There was a notable increase in the detection of non-fermenting bacteria during the period of 2022–2023, especially represented by *P.aeruginosa* and *S.maltophilia*. The ratio of Gram-negative bacteria to Gram-positive bacteria exhibited a gradual increase across quarters.These variations may be influenced by alterations in temperature and humidity affecting bacterial growth, as well as by selective pressures resulting from antibiotic usage during certain periods. Our analysis of the composition of bacterial strains in the early and late-onset POP showed no significant differences. These findings strongly accentuate the imperative for clinicians to consider the annual and seasonal variations in pathogen distribution when selecting empirical antimicrobial therapy.

The circumstance of MDRO in healthcare-associated infections becomes progressively severe, while the available research on the antibiotic resistances of POP-causing pathogens in CABG patients was limited. Our study revealed that the resistance patterns of common POP pathogens were concerning: *A.baumannii* exhibited the highest resistance rates, followed by *K.pneumoniae* and *P.aeruginosas*,with resistance rates to commonly used antibiotics being > 50%, < 50%, and < 20%,respectively. *S.aureus* showed resistance rates above 40% to several

**Table 7. Comparative analysis of disease burden-related indicators in different groups of postoperative pneumonia patients.**

| Indicator | POP Group (n = 518) | non-POP Group (n = 4051) | P value * |
|---|---|---|---|
| ICU stay,d,median (IQR) | 9.00 (5.00, 14.00) | 4.00 (3.00, 11.75) | 0.002 |
| Postoperative hospital stay,d,median (IQR) | 20.00 (13.00, 31.25) | 15.50 (10.25, 19.75) | <0.001 |
| Hospitalization costs, ¥ ,median (IQR) | 255592.15 (193384.29, 336337.53) | 180501.02 (154493.58, 220501.03) | <0.001 |
| **Indicator** | **MDRO-POP Group (n = 228)** | **non-MDRO-POP Group (n = 290)** | **P value *** |
| ICU stay,d,median (IQR) | 11.00 (7.00, 16.00) | 7.00 (3.00, 9.00) | <0.001 |
| Severe pneumonia (n) | 44 (19.30) | 16 (5.51) | <0.001 |
| Postoperative hospital stay,d,median (IQR) | 22.00 (13.00, 34.00) | 17.00 (12.00, 25.00) | 0.021 |
| Hospitalization costs, ¥ ,median (IQR) | 273137.78 (204265.65, 337002.90) | 224766.03 (185254.94, 315148.71) | <0.001 |
| **Indicator** | **Simple infection Group (n = 483)** | **Co-infections Group (n = 35)** | **P value *** |
| ICU stay,d,median (IQR) | 9.00 (4.00, 14.00) | 10.00 (7.00,16.00) | 0.189 |
| Severe pneumonia (n) | 46 (9.52) | 14 (40.00) | <0.001 |
| Postoperative hospital stay,d,median (IQR) | 20.00 (13.00, 31.00) | 26.00 (16.00, 35.00) | <0.001 |
| Hospitalization costs, ¥ ,median (IQR) | 242174.39 (191320.92, 336115.74) | 285400.82 (255684.40, 330173.81) | <0.001 |

*P values were calculated using the Mann-Whitney U test.

antibiotics. Compared with the period of 2020–2021, a general elevation in the resistance rates of *K.pneumoniae* to commonly utilized antibiotics were observed in 2022 - 2023, with notable increases for ciprofloxacin and ceftriaxone (both > 10%). Similarly, *P.aeruginosa* showed significant increases in resistance to ticarcillin/clavulanate and aztreonam (both > 10%). Conversely, *A.baumannii* showed a decreasing trend in resistance to most antibiotics. The resistance rates of *K.pneumoniae* to imipenem and meropenem manifested an upward trend, while *P.aeruginosa* and *A.baumannii* exhibited slight declines. *S. aureus* demonstrated reduced resistance rates to most antibiotics, yet the resistance rates to trimethoprim-sulfamethoxazole and levofloxacin persisted above 20%. The emergence of resistance to polymyxins in *K.pneumoniae* and tigecycline in *A.baumannii* in 2022–2023 warrants attention. Clinicians ought to contemplate these evolving resistance patterns when making the selection of empirical antibiotics to evade highly resistant drugs.

Research conducted by Ren J et al. [18]discovered that the detection rate of MDRO among causing POP-causing pathogens in cardiac surgery was 48.6%, and 53.7% for all hospital infections related to CABG. Our study found a marginally higher MDRO detection rate of 57.7% in patients developed POP after CABG.Moreover, the detection rate of MDRO was notably higher in Gram-positive bacteria (82.1%) than in Gram-negative bacteria (53.9%), highlighting the substantial challenge of Gram-positive bacteria's multi-drug resistance to clinical therapeutics and its potential impact on patient outcomes. Arvanitis M et al. [19] implied that there was a significantly higher detection rate of MDRO in late-onset ventilator-associated pneumonia. Similarly,in our study, we discovered that the late-onset POP was more frequently associated with MDRO infection, as represented by MDR-Enterobacteriaceae, potentially reflecting the selective pressure exerted by antibiotic usage in postoperative care.

Greco G et al. [1] reported that postoperative infections in cardiac surgery patients result in an extended hospital stay of 2 weeks and increased costs by nearly $38,000. Thompson MP et al. [20] found that POP following CABG was associated with longer hospital stays (4.1 days) and increased costs by 24.5% ($46,723 vs. $37,496), both statistically significant (P < 0.001). In our study, CABG patients with POP experienced a 5.00-day increase in median ICU stay,a 4.50-day increase in median postoperative hospital stay, and an increase in median hospitalization cost of ￥75091.13 (P < 0.001). Xu B et al. [21] found that patients infected

with MDRO had an average increase in hospital stay of 4.11-day compared to non-MDRO infected patients (P < 0.001). A meta-analysis of the direct economic burden of nosocomial infection caused by MDRO across multiple countries reported average hospital stay extensions and direct economic losses ranging from $916.61 to $98,575 [22]. Another meta-analysis focusing on ICU patients with MDRO infection in China found that the average hospital costs were ￥96,450 higher than those without MDRO infection (P < 0.05) [23]. Our study was consistent with previous findings by demonstrating that MDR infection significantly elevated the risk of severe pneumonia and were associated with extended lengths of stay in the length of ICU stay and postoperative hospital stay, as well as increased hospitalization costs (all P < 0.05). Furthermore, patients with co-infection were also at a higher risk of severe pneumonia, with consequent prolongation of postoperative hospital stay and escalation of hospitalization costs (all P < 0.001).There were few literatures directly adressing the disease burden of bacteria co-infections in patients undergoing surgery, yet the results are explicable. Co-infections complicate antimicrobial therapy due to the increased difficulty in selecting effective agents, potentially exacerbating endogenous flora imbalance through new acquisitions or prolonged antibiotics use, thereby leading to more pronounced discrepancies in postoperative hospital stay and hospitalization costs.

## Conclusion

Our findings suggest high overall incidence of POP following CABG, with observed fluctuations across different years. The occurrence of POP,along with the prevalent pathogens responsible for it and their resistance profiles might exhibit dynamic fluctuations, underscoring the necessity for continuous regional surveillance to inform and refine empirical anti-infective therapeutics. This study acknowledges several limitations due to its retrospective and single-center design. Firstly, incomplete extraction of historical antimicrobial susceptibility data could potentially introduce information bias. Secondly,the stringent case selection criteria employed to control for confounding factors may limit the generalizability of our findings. This suggests that further studies are needed to determine the extent to which these results can be applied to broader populations. Future research will focus on elucidating the mechanisms underlying the evolution of pathogen spectra, as well as developing and evaluating antibiotic stewardship strategies. Conducting multicenter, large-sample studies will be essential to underpin evidence-based empirical prescribing, enhancing early treatment success rates, and mitigating the emergence and spread of resistance among pathogens.

## Supporting information

**S1 File. Data for analysis.**
(XLSX)

## Author contributions

**Conceptualization:** Jian Zhang, Yang Yang, Guojun Zhang.

**Data curation:** Yuxiao Zhan.

**Formal analysis:** Yuxiao Zhan, Jian Zhang.

**Funding acquisition:** Jian Zhang.

**Investigation:** Yuxiao Zhan, Jian Zhang, Yang Yang.

**Methodology:** Jian Zhang, Yang Yang, Rui Yang, Guojun Zhang.

**Project administration:** Guojun Zhang.

**Resources:** Guojun Zhang.

**Supervision:** Rui Yang, Guojun Zhang.

**Validation:** Jian Zhang, Yang Yang, Rui Yang.

**Writing – original draft:** Yuxiao Zhan.

**Writing – review & editing:** Yuxiao Zhan.

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
