## [Decision Letter · Decision Letter 0]

31 Oct 2024

PONE-D-24-34876Postoperative Pneumonia in Isolated Coronary Artery Bypass Grafting: A Comprehensive Study of Epidemiology, Etiology, and Disease Burden from China (2020-2023)PLOS ONE

Dear Dr. Zhang,

Thank you for submitting your manuscript to PLOS ONE. After careful consideration, we feel that it has merit but does not fully meet PLOS ONE’s publication criteria as it currently stands. Therefore, we invite you to submit a revised version of the manuscript that addresses the points raised during the review process.

We look forward to receiving your revised manuscript.

Kind regards,

Mohamed O Ahmed, Ph.D

Academic Editor

PLOS ONE

Journal requirements: When submitting your revision, we need you to address these additional requirements. 1. Please ensure that your manuscript meets PLOS ONE's style requirements, including those for file naming. The PLOS ONE style templates can be found at https://journals.plos.org/plosone/s/file?id=wjVg/PLOSOne_formatting_sample_main_body.pdf and https://journals.plos.org/plosone/s/file?id=ba62/PLOSOne_formatting_sample_title_authors_affiliations.pdf 2. We note that the grant information you provided in the ‘Funding Information’ and ‘Financial Disclosure’ sections do not match.  When you resubmit, please ensure that you provide the correct grant numbers for the awards you received for your study in the ‘Funding Information’ section. 3. We note that your Data Availability Statement is currently as follows: [All relevant data are within the manuscript and its Supporting Information files.] Please confirm at this time whether or not your submission contains all raw data required to replicate the results of your study. Authors must share the “minimal data set” for their submission. PLOS defines the minimal data set to consist of the data required to replicate all study findings reported in the article, as well as related metadata and methods (https://journals.plos.org/plosone/s/data-availability#loc-minimal-data-set-definition). For example, authors should submit the following data: - The values behind the means, standard deviations and other measures reported;- The values used to build graphs;- The points extracted from images for analysis. Authors do not need to submit their entire data set if only a portion of the data was used in the reported study. If your submission does not contain these data, please either upload them as Supporting Information files or deposit them to a stable, public repository and provide us with the relevant URLs, DOIs, or accession numbers. For a list of recommended repositories, please see https://journals.plos.org/plosone/s/recommended-repositories. If there are ethical or legal restrictions on sharing a de-identified data set, please explain them in detail (e.g., data contain potentially sensitive information, data are owned by a third-party organization, etc.) and who has imposed them (e.g., an ethics committee). Please also provide contact information for a data access committee, ethics committee, or other institutional body to which data requests may be sent. If data are owned by a third party, please indicate how others may request data access.

Reviewers' comments:

Reviewer's Responses to Questions

**Comments to the Author**

1. Is the manuscript technically sound, and do the data support the conclusions?

Reviewer #1: Yes

Reviewer #2: Yes

2. Has the statistical analysis been performed appropriately and rigorously? 

Reviewer #1: I Don't Know

Reviewer #2: No

3. Have the authors made all data underlying the findings in their manuscript fully available?

Reviewer #1: Yes

Reviewer #2: No

4. Is the manuscript presented in an intelligible fashion and written in standard English?

Reviewer #1: Yes

Reviewer #2: No

5. Review Comments to the Author

Reviewer #1: dear authors your topic is very interesting. Your manuscript needs some grammatically revisions.

Please use high quality figures in your manuscript.

overally this manuscript can be acceptable after minor revision.

Reviewer #2: The article entitled "Postoperative Pneumonia in Isolated Coronary Artery Bypass Grafting: A Comprehensive Study of Epidemiology, Etiology, and Disease Burden from China (2020-2023).”

is generally well-written and logically structured. However, some sections could benefit from more concise language to improve clarity.

This study contributes significantly to the understanding of postoperative pneumonia in CABG patients, particularly regarding the epidemiology and resistance patterns of pathogens in a Chinese context.

The title is clear and informative.

Abstract: remove the statistical part of the method. It is deficient and doesn’t present the actual tests used.

The results section is narrative. Please support this with numbers and exact p-values.

Introduction

Please revise the objectives. The aim is not to furnish the scientific evidence. This could be a secondary and indirect objective.

Methods

Write in paragraphs with complete sentences, not bullets.

Please indicate how confounders were controlled.

Results

Use subsection to facilitate reading.

In the tables, don’t mention levels of mutually exclusive variables.

Present non-normal data as median and IQR.

Discussion

Well-writen

Please expand the limitations and potential biases.

Elaborate on area of further research.

6. PLOS authors have the option to publish the peer review history of their article (what does this mean?). If published, this will include your full peer review and any attached files.

Reviewer #1: No

Reviewer #2: **Yes: **Amr Arafat

---

## [Author Response · Author response to Decision Letter 1]

2 Dec 2024

Dear Editor and reviewers,

We are grateful for the thorough and constructive feedback provided on our manuscript entitled "Postoperative Pneumonia in Isolated Coronary Artery Bypass Grafting: A Comprehensive Study of Epidemiology, Etiology, and Disease Burden from China (2020-2023)". Your insights have been invaluable in enhancing the quality and clarity of our work. We have meticulously addressed each of your comments and thoroughly reviewed and checked the entire manuscript content, making the following revisions:

Response to academic editor:

1.Please ensure that your manuscript meets PLOS ONE's style requirements, including those for file naming.

Thank you for your guidance.We have carefully reviewed the submission guidelines and the formatting standards of recently published articles of PLOS ONE, and have made the necessary adjustments to our submission. Specifically, regarding file naming, we have renamed the figure files according to the journal's specifications to ensure clarity and consistency. We have also double-checked all other aspects of our manuscript to comply with the required formatting, referencing, and submission standards.

2.The grant information provided in the ‘Funding Information’ and ‘Financial Disclosure’ sections do not match. When you resubmit, please ensure that you provide the correct grant numbers for the awards you received for your study in the ‘Funding Information’ section.

Thank you for bringing this discrepancy to our attention. We have reported the funding information in the 'Funding Information' section, but we did not make the correct disclosure statement in the 'Financial Disclosure' section.We apologize for the oversight and appreciate your guidance on maintaining accuracy in our submission.Upon reviewing the 'Funding Information' and 'Financial Disclosure' sections, we have identified the mismatch in the grant information provided. We understand the importance of consistency and accuracy in these sections, especially as they pertain to the integrity of the research and transparency in funding.We have taken the necessary steps to correct this issue. The grant numbers associated with our study have been verified and will be accurately reflected in the 'Funding Information' section upon resubmission. We will ensure that the grant information is consistent across all relevant sections of the manuscript to avoid any further confusion.

3. We note that your Data Availability Statement is currently as follows: [All relevant data are within the manuscript and its Supporting Information files.] Please confirm at this time whether or not your submission contains all raw data required to replicate the results of your study.

Upon your request, we would like to clarify that the initial submission's omission of comprehensive raw data was attributable to a preliminary misunderstanding regarding the necessity of submitting such data.We have provided all the raw data necessary to replicate the findings of the study, in accordance with the "minimum data set" requirements. This additional data is now included in its Supporting Information files.The updated submission ensures that all necessary data are available for peer review and for any future researchers who may wish to validate our findings.We have taken the necessary steps to correct this oversight and appreciate your guidance in maintaining the standards of scientific rigor. The revised submissions now fully complies with the journal's policies on data sharing and availability.

4.While revising your submission, please upload your figure files to the Preflight Analysis and Conversion Engine (PACE) digital diagnostic tool. PACE helps ensure that figures meet PLOS requirements

As per your instructions, we have renamed our figure files to meet the PLOS ONE requirements and have uploaded them to the Preflight Analysis and Conversion Engine (PACE) digital diagnostic tool.We have ensured t hat all figures comply with the PLOS ONE guidelines, and we have received confirmation from PACE that our figures are now in compliance.We believe that these revisions have significantly improved the quality and clarity of our manuscript, and we are confident that it now meets the high standards of your esteemed journal.

Response to reviewers:

Reviewer #1:

Your manuscript needs some grammatically revisions.Please use high quality figures in your manuscript.

Thank you for your feedback on our manuscript.We have taken the comments seriously and have made the following revisions:

(1)Grammatical Revisions

We have conducted a thorough review of the entire manuscript for grammatical accuracy. With the assistance of a professional with extensive experience in academic writing, we have corrected all grammatical errors and reorganized the language to align with the common expression habits suitable for journal publication. This process has ensured that our manuscript is now more polished and professional.

(2)High-Quality Figures

In response to the comment regarding the quality of figures, we have carefully reviewed the PLOS ONE guidelines for figures and recent articles to ensure our figures meet the highest standards. We have revised our figures accordingly and are confident that they are now of the quality expected by the journal.

We believe these revisions have significantly improved the manuscript and address the concerns raised by the reviewers. We are grateful for the opportunity to enhance our work and hope that it is now acceptable for publication in PLOS ONE.

Reviewer #2:

Thank you for your thorough review and valuable feedback on our manuscript. We have carefully considered each point and have made the following detailed revisions:

1.Some sections could benefit from more concise language to improve clarity.

We appreciate your suggestion to enhance the clarity of our manuscript through more concise language. To address this, we have taken the following steps:

(1)Engaged a professional with extensive experience in academic writing to review and revise the manuscript.(2)Identified and streamlined lengthy sentences and paragraphs to ensure that each point is conveyed as succinctly as possible without sacrificing detail or accuracy.(3)Reorganized complex sections to improve the flow of information and enhance readability.(4)Replaced jargon with more accessible language where appropriate, ensuring that the manuscript is accessible to a broad readership.(5)Conducted multiple rounds of editing to ensure that every section of the manuscript is clear, concise, and adheres to the highest standards of scientific communication.We believe these revisions have significantly improved the manuscript's clarity and readability, aligning with the expectations of PLOS ONE and its readership.

2.Abstract: (1)Remove the statistical part of the method. It is deficient and doesn’t present the actual tests used.(2)The results section is narrative. Please support this with numbers and exact p-values.

We have carefully considered your comments on the abstract section and have made the following revisions in response to your specific instructions:

(1)Statistical Method Removal

We understand your concern regarding the deficient statistical part in the abstract's method section. In response, we have removed the statistical details that were previously included. We have also optimized the language to ensure clarity and focus on the core methods used in our study.

(2)Supporting Data in Results Section

You correctly pointed out the need for numerical support in the results section of the abstract. In response to your suggestion, we have added specific numerical values for the key results in the abstract's results section, explained the data supporting these key results,and provided exact p-values to support the statistical significance of our results. We have ensured that the presentation of the results is clear, orderly, and coordinated with the methods and conclusions sections. Through these specific improvements, we believe that the results section of the abstract is now more precise and persuasive, better reflecting the depth and quality of our research.

Additionally, we identified that the original conclusion in the abstract did not optimally articulate the true implications of our research findings. To enhance the clarity and transparency of our conclusions, we have conducted a thorough reassessment and restructured the conclusion to more directly convey the principal findings of our study. We have ensured that the revised conclusion closely aligns with the objectives and outcomes of our research.We hope that these revisions meet your requirements and enhance the informational value of the abstract.

3.Introduction:Please revise the objectives. The aim is not to furnish the scientific evidence. This could be a secondary and indirect objective.

We have revised the introduction to ensure that the primary objectives of our study are clearly stated and prioritized. The aim of our research is now explicitly defined as [To delineate the epidemiological patterns and etiological profiles of POP in the CABG population, as well as to quantify its impact on disease burden], which is a direct and primary goal rather than a secondary or indirect one.We hope this study will guide the early initiation of appropriate empirical anti-infective protocols in clinical practice and highlight the necessity of close surveillance for POP.We aim to achieve the prevention and control of POP ultimately, through our targeted research efforts and clinical guidance.

To further enhance the clarity of our objectives, we have restructured the introduction. We have added emphasis to the direct goals of our study, ensuring that the scientific community understands the immediate and practical implications of our work. This focus on direct objectives aligns with the primary intent of our research and provides a clear direction for the study.

4.Methods:(1)Write in paragraphs with complete sentences, not bullets.(2)Please indicate how confounders were controlled.

(1)Reformatting to Paragraphs

We have taken your comments to heart and have made substantial revisions to enhance the clarity and rigor of our methods description.In accordance with PLOS ONE's guidelines and recent publication standards, we have converted the methods section from a bullet-point format to full sentences and paragraphs. This narrative style not only aligns with the journal's requirements but also provides a more coherent and reader-friendly explanation of our methodology.We have paid particular attention to the sections detailing the exclusion criteria and diagnostic standards. These have been rewritten in complete sentences and structured into paragraphs to enhance clarity and readability. This reformatting ensures that the criteria are explicitly stated, facilitating a clear understanding of the study's participant selection process.

(2)Control of Confounding Factors

Recognizing the critical role of controlling confounders in ensuring the validity of our study's findings, we have expanded the methods section to include a detailed description of the specific measures we employed:a)Using strict inclusion and exclusion criteria;b)Handling of missing values;c)multivariate regression analysis to adjust for potential confounders.

(3)Statistical Analysis Enhancements

We have augmented the statistical analysis section with a clear explanation of the procedures used to assess the normality of the distribution of quantitative data. This includes:[Specify the test used, as "the Kolmogorov-Smirnov test for normality"].Additionally, we have detailed the statistical methods actually used in the study, ensuring that the description accurately reflects the analyses conducted. Adjustments have been made to align the methods with the specific statistical tests applied.

By implementing these revisions, we have ensured that our methods section is not only in compliance with PLOS ONE's standards but also provides a transparent and comprehensive account of how our study was conducted. We hope these changes will address your concerns and contribute to the overall quality and transparency of our research report.

5.Results:(1)Use subsection to facilitate reading.(2)In the tables, don’t mention levels of mutually exclusive variables.(3)Present non-normal data as median and IQR.

In response to your feedback, we have made the following revisions and additions:

(1)Addition of Subheadings

We have incorporated subheadings throughout the results section to enhance the readability and logical flow of the content. This is particularly beneficial in sections with a large volume of data, such as the disease burden, where subheadings help readers to more easily follow the different parts of the results.

(2)Treatment of Mutually Exclusive Variables in Tables

We have carefully reviewed all tables to ensure that the levels of mutually exclusive variables are not mentioned unless they are critical to the analysis. Regarding the mutually exclusive variable grading mentioned in Tables 2-4, we would like to provide the following explanation and seek your agreement. We have subcategorized the strains based on Gram staining and whether they are intestinal fermenting bacteria to systematically analyze the distribution characteristics and resistance patterns of pathogens. The top five common pathogens of postoperative pneumonia account for a significant proportion, but the distribution of the remaining strains is scattered, and due to their low numbers,the significance of separate analysis may not be substantial, thus not presented in the tables. However, in routine monitoring, we have observed that these uncommonly detected strains may alter the distribution trends within their subcategories, leading to statistically significant differences. We hope that our statistical analysis, categorized by the affiliation and classification of pathogens, will provide clinical insights. To facilitate reading and ensure that the presentation of this information does not confuse readers, we have aligned the classification levels of the pathogen strains in a columnar manner to improve the clarity and readability of the tables.

(3)Description of Non-Normal Distribution Data

We have re-examined the quantitative data involved in the article for normality and.Based on the assessments, we have opted to use the median (IQR) as descriptive statistical measures for non-normal data. Corresponding adjustments have been made in the text and tables.

We anticipate that these modifications will better meet the standards of academic publishing and enhance the presentation of our research findings. We appreciate your meticulous review and valuable suggestions.

6.Discussion:(1)Please expand the limitations and potential biases.(2)Elaborate on area of further research.

(1)Expansion on Limitations and Potential Biases

We have expanded the limitations section at the end of our article to provide a more comprehensive understanding of the constraints and potential biases that may have influenced our study. We have carefully examined the methods and results to identify areas where our findings may be limited, such as sample size restrictions, potential selection biases, or limitations in the generalizability of our results. We have also discussed how these limitations might affect the interpretation of our findings and their implications for future research.

(3)Elaboration on Areas for Further Research

In response to your request, we have supplemented the discussion section of our article with a detailed exposition of the limitations and potential biases of this study, and have further indicated directions for future research. We have delineated specific avenues for future studies. We have id

---

## [Decision Letter · Decision Letter 1]

27 Jan 2025

Postoperative Pneumonia in Isolated Coronary Artery Bypass Grafting: A Comprehensive Study of Epidemiology, Etiology, and Disease Burden from China (2020-2023)

PONE-D-24-34876R1

Dear Dr.Zhang,

We’re pleased to inform you that your manuscript has been judged scientifically suitable for publication and will be formally accepted for publication once it meets all outstanding technical requirements.

Kind regards,

Mohamed O Ahmed, Ph.D

Academic Editor

PLOS ONE

---

## [Editor Report · Acceptance letter]

PONE-D-24-34876R1

PLOS ONE

Dear Dr. Zhang,

I'm pleased to inform you that your manuscript has been deemed suitable for publication in PLOS ONE. Congratulations! Your manuscript is now being handed over to our production team.

Kind regards,

on behalf of

Dr. Mohamed O Ahmed

Academic Editor

PLOS ONE